# Adequate Management of Postoperative Complications after Esophagectomy: A Cornerstone for a Positive Outcome

**DOI:** 10.3390/cancers14225556

**Published:** 2022-11-12

**Authors:** Imad Kamaleddine, Alexander Hendricks, Magdalena Popova, Clemens Schafmayer

**Affiliations:** Department of General, Visceral, Vascular, Thoracic and Transplantation Surgery, Rostock University Hospital, Schillingallee 35, 18057 Rostock, Germany

**Keywords:** esophageal cancer, postoperative complications, complication management, endoscopy, anastomosis insufficiency, esophago-respiratory fistula, chylothorax

## Abstract

**Simple Summary:**

Despite advances in the multimodal therapy of esophageal cancer, postoperative complications remain a challenge for surgeons and patients. Their prompt diagnosis and adequate treatment are essential for the further course of cancer therapy, ensuring an optimal outcome. The tumor biology was not found to be associated with a better or worse outcome when it comes to postoperative complications. In this review, we focus on up-to-date approaches to prevent and treat the major complications after esophagectomy for cancer. Among those are the anastomotic leak (AL), the esophago-respiratory fistula (ERF), and the chylothorax (CT).

**Abstract:**

Background: Esophagectomy for cancer is one of the most complex procedures in visceral surgery. Postoperative complications negatively affect the patient’s overall survival. They are not influenced by the histology type (adenocarcinoma (AC)/squamous cell carcinoma (SCC)), or the surgical approach (open, laparoscopic, or robotic-assisted). Among those dreadful complications are anastomotic leak (AL), esophago-respiratory fistula (ERF), and chylothorax (CT). Methods: In this review, we summarize the methods to avoid these complications, the diagnostic approach, and new therapeutic strategies. Results: In the last 20 years, both centralization of the medical care, and the development of endoscopy and radiology have positively influenced the management of postoperative complications. For the purpose of their prevention, perioperative measures have been applied. The treatment includes conservative, endoscopic, and surgical approaches. Conclusions: Post-esophagectomy complications are common. Prevention measures should be known. Early recognition and adequate treatment of these complications save lives and lead to better outcomes.

## 1. Introduction

The incidence of esophageal adenocarcinoma (AC) is increasing in high-income countries, with excess body weight, gastroesophageal reflux, and Barrett’s esophagus being the main risk factors. Esophageal cancer, with histological subtypes (AC) and squamous cell carcinoma (SCC), ranks sixth on the worldwide cancer-related deaths scale [1]. On the other hand, esophagectomy, one of the most complex surgeries for cancer, remains the only curative treatment, despite advanced multimodal therapies [2]. Esophagectomy is associated with a high complication rate reaching 59%; among those, 17.2% are defined as major complications with a Clavian–Dindo grade greater than IIIB and a 90-day mortality rate of 4.5% [3,4]. The risks for post-esophagectomy complications are multivariable. The patient’s co-morbidities play an important role, with the most negatively influential being heart and lung diseases [5], diabetes [6], morbid obesity [7], associated malnutrition [8], and smoking [9]. A new systematic review and meta-analysis of the preoperative physical fitness of patients showed that an unfit status was associated with a higher risk of postoperative morbidity and mortality in patients with esophageal cancer [10]. Recent reviews of postoperative complication management suggested that the surgical team should have a low threshold for scrutinizing their occurrence [11,12,13]. The golden rule is to investigate any deviation from the normal postoperative course as delays in diagnosis or treatment timing can increase the severity of complications, as seen in Figure 1. As surgeons, we want to focus on the most feared surgical-related complications, such as anastomotic leak (AL), esophago-respiratory fistula (ERF), and chylothorax (CT) in order to discuss the prevention of their occurrence, early recognition, and successful treatment.

## 2. Anastomotic Leak

Blencowe et al. remarked in their systematic review of clinical outcomes after esophagectomy that 63% of the included articles lacked a definition of the AL [11]. This led the Esophageal Complications Consensus Group (ECCG) in 2015 to define AL as a full-thickness gastrointestinal defect involving the esophagus, anastomosis, staple line, or conduit, irrespective of presentation or method of identification [12]. It is divided into three types depending on its treatment. Type 1 is medically or conservatively treated, Type 2 needs intervention but not surgical therapy, and Type 3 requires surgical intervention. The AL complicates 2–15.9% of the cases [13]. AL can be cervically or intrathoracically localized depending on the performed operation. A recent meta-analysis concluded that the type of anastomosis (hand-sewn or stapled) did not show any superiority when it comes to AL [14]. Nowadays, the standard of the surgical approach includes esophageal resection, two-field lymph node dissection, building of a neo-esophagus from a stomach conduit, cholecystectomy, and insertion of a jejunostomy for nutrition purposes. The lack of a serosa, which makes the tissues more fragile, and the specific vascularization of the anastomosis make the esophageal anastomosis more susceptible to leaks compared to any other gastrointestinal anastomosis. The neo-esophagus solely relies on the blood supply from the right gastroepiploic artery and the submucosal network in the remained esophagus. The longer the gastric conduit is, the more susceptible it will be to ischemic changes. Therefore, the cervical anastomosis is riskier than the other type with a leak rate from 25–45% compared to 5–15% with intrathoracic anastomosis [15].

### 2.1. Prevention

No complication can be 100% prevented, but some measures should be applied to minimize its risk, as outlined below.

#### 2.1.1. Preoperatively

Even though most patients will present some sort of malnutrition, preoperative optimization of the nutritional status can be always helpful [16].

#### 2.1.2. Intraoperatively

The preparation of the gastric conduit should be meticulous along the great curvature, tissues should be gently handled and care should be taken to build a 4 cm width conduit in order to not affect the submucosal vascularization, leading to a conduit tip necrosis and a subsequent leak.

#### 2.1.3. Postoperatively

The recommendations for enhanced recovery include resuming enteral nutrition as soon as possible, promoting early mobilization and pulmonary physiotherapy, and ensuring good tissue perfusion by maintaining normal cardiac output [17].

### 2.2. Diagnosis

Early diagnosis is essential for the success of the treatment. Any deviation from the norm in the postoperative period needs be investigated to exclude AL or to promptly treat it once confirmed. Although the majority of patients present classic signs of infection such as elevated C-reactive protein (CRP), procalcitonin and/or leukocytosis, tachycardia and fever, or an abnormal drainage discharge (saliva or bile), one should not overlook a new onset of atrial fibrillation or postoperative delirium [18]. The following diagnostic tools/methods are used: endoscopic intervention [19], esophagogram using water-soluble contrast, computer tomography (CT-Scan) with oral (oc) and intravenous (iv) contrast, or oral uptake of methylene blue [20].

### 2.3. Therapy

Some general principles should be followed: adequate drainage and sepsis control while taking into consideration the patient’s current hemodynamic status, anastomotic localization (cervical or intrathoracic), and AL size. There is no general consensus for the treatment of AL, and this largely depends on the surgical team’s experience and preference. The treatment can be divided into three groups: conservative, endoscopic, and surgical. Sometimes, a combination of these therapies is mandatory.

#### 2.3.1. Conservative

Conservative therapy consists of withholding oral feeding while ensuring nutritional support, preferably enteral, through an intraoperatively placed jejunostomy [13,15,19,20,21]. An alternative is parenteral feeding through a central venous line. To control systemic inflammatory reaction, an intravenous wide-spectrum antibiotic with an antifungal should be started as soon as possible. As previously mentioned, a cervical anastomosis is more likely to lead to a leak, which is easily accessible for local treatment by simply opening the skin and draining the wound. Its risk of aggravation to a mediastinitis and associated mortality is lower when compared to an intrathoracically anastomosis [15,22].

#### 2.3.2. Endoscopic

In the last two decades, In the last two decades, the management of AL after esophagectomy benefited from development in the endoscopic field.. This approach gained popularity because it is diagnostic and therapeutic at the same time. To obtain the best results, an advanced and all-around-the-clock endoscopic experience is needed. Because of this, specialized centers prove to have better outcomes, when compared to smaller hospitals [23]. AL usually occurs in the second postoperative week [20]. An endoscopy under sedation around that time is usually safe, feasible [24], and very crucial to assess the local finding: size and location of the AL, and tissue vascularization at the anastomotic site or the gastric conduit. The endoscopic approach offers a variety of treatment options such as using fibrin glue and plugs, clipping, the use of either the Through-The-Scope-Clips (TTSC) or Over-The-Scope-Clips (OTSC^®^) (Ovesco Endoscopy GmbH, Tübingen, Germany), the use of self-expanded metal stent (SEMS) [25], endoscopic vacuum therapy (EVT) [26], and endoscopic suturing. After an initial insertion of the fibrin adhesive, a vicryl plug is administered and is mainly suitable for minor leaks with minimal surrounding inflammation. However, in several cases, this should be followed by either surgical intervention or stent application [27]. TTS Clips, invented in 1974 for hemostasis, were also successfully used to treat perforations and AL. They are applied through the working channel [28]. Fully opened TTS clips ready for application span 11 mm and are hence optimal for defects <10 mm in size. Multiple Clips can be fired and placed next to each other [29]. For larger defects (10–20 mm), OTSC^®^ are more useful [30]. Invented in 2007, OTSC^®^ consist of an over-the-scope applicator cap with a mounted nitinol clip. The clip itself is fitted with anchors that safely and appropriately compress the tissue. The tissue will be fully loaded into the cap at the distal end of the endoscope either by suction or with the help of a grasper device. The clip is then released by manual force [31]. SEMSs, which were primarily introduced in the 1990s to treat dysphagia and esophagus stenosis, are available as metallic, plastic, covered, and non-covered stent [32]. They are composed from either Elgiloy (cobalt, nickel, and chrome) or Nitinol (nickel and titanium). Covered stents are coated with either a polymer such as silicon, poly-urethane, or other biomaterials. In the case of an AL with a larger defect than what a clip can treat, SEMS with adequate CT-Scan-guided drainage of the paraesophageal hole can close the defect and ensure oral feeding. Following an easy placement, the SEMS usually needs up to 24 h to fully expand [33]. On the other hand, some teams such as ours are more prone to use the EVT as a first step approach. In this case, an open-pore poly-urethane sponge attached to a drainage tube is inserted in the cavity and connected to a vacuum pump, creating negative pressure. Thus, it drains, cleans, and decreases the cavity volume by promoting wound healing through enhancing the formation of granulation tissue. This procedure can be performed under sedation and as a bedside exam. The maneuver should be repeated every 3–4 days. The sponge should be changed and adjusted accordingly. Usually, the first sponge placement is followed by a CT-Scan with iv contrast, as shown in Figure 2, to rule out the presence of an abscess behind the AL or a mediastinitis. Endoscopically, even if the AL is not confirmed or just measures a few mms, an endoluminal sponge can be easily placed. In a few days, another endoscopy will be performed and the suspected area of the AL will be re-evaluated. Here, we face three scenarios:

If there is no leak, the sponge can be safely removed.

If the leak’s width is still a few mms wide, but an insufficiency hole is seen on the CT-Scan, the leak will be endoscopically balloon-dilated and a sponge will be placed in the hole.

If a paraesophageal extraluminal insufficiency hole is already seen, the sponge will be extraluminally placed.

Endoscopic suturing methods have also been developed, but their clinical application in treating esophageal leaks is limited. Case reports have been published describing the use of OverStitch Endoscopic Suturing System (Apollo Endosurgery, Austin, TX, USA). The system’s needle driver attaches to the distal end of an endoscope, whereas the OverStitch handle that triggers the needle driver is placed at the proximal end of the endoscope, allowing for endoluminal suturing. Both absorbable and non-absorbable threads can be used, and suturing can be performed either by a single stitch or continuously [34]. Advanced endoscopic experience is essential as a combination of more than one approach is sometimes needed, for example, primarily treating the AL with EVT, followed by the placement of a SEMS with fixation using an OTSC^®^ or even suturing.

#### 2.3.3. Surgical

Because of the high mortality associated with re-intervention after esophagectomy [35], this option is indicated in the following situations: when a leak is evident in the first 72 h postoperatively, when ischemia of the gastric conduit is diagnosed endoscopically or as salvage therapy, and when all other conservative and endoscopic approaches fail [36]. However, the mortality rate after a surgical intervention is higher compared to patients treated endoscopically [37]. The surgery aims to eliminate the septic source, to drain the mediastinum, and to reinforce the anastomotic defect either by suturing or by using a muscle flap repair. The anatomical structures used to build a flap are the diaphragm, latissimus dorsi, serratus anterior, or one of the pectoralis muscles depending on the anastomotic location [38]. Sometimes, the anastomosis should be re-built. This should be in a tension-free manner. If this is not possible, an esophageal diversion with a cervical end-esophageal exteriorization should be performed. Esophageal surgeons should be familiar with this technique because it is considered to be the last resort approach [39]. The reconstruction operation should be performed once the infection subsides, the patient is stabilized, and the nutritional status is optimized. This can be performed using the stomach or the colon. The perfect timing is still controversial, but it is rarely performed earlier than six months after the esophagus diversion surgery [40].

## 3. Esophago-Respiratory Fistula

An ERF represents an abnormal connection between the digestive and the respiratory system. It is classified into the esophago-tracheal fistula (ETF); esophago-bronchial fistula (EBF); and more rarely, esophago-pulmonary fistula (EPF) depending on its location and whether it is benign or malignant depending on its etiology [41]. The post-esophagectomy ERF is usually caused by a direct iatrogenic airway membranous injury during the surgery or as a consequence of a non-drained intrathoracic leak [42]. It is rare, representing around 1% of the complications after esophagectomy for cancer and associated with a high mortality rate reaching 38.5% [43].

### 3.1. Diagnosis

The symptoms range from a nonspecific cough to severe sepsis associated with mediastinitis, with onset almost always after oral intake. Diagnosis is either endoscopically done (simultaneous bronchoscopy and gastroscopy) after injecting methylene blue dye through the gastroscope and witnessing its presence in the bronchial tree [44] or by CT-scan with (oc) and observing the contrast in the bronchus or the lower pulmonary lobe [45].

### 3.2. Therapy

The treatment’s success depends on the fistula type, its width, the patient’s pulmonary status, and the center’s experience. While treatment of AL is well-described using endoscopic methods, research of the literature just found one national expert consensus on diagnosis and management of the ERF [46]. Reported treatment options can be endoscopic or surgical.

#### 3.2.1. Endoscopic

Among the described endoscopic approaches are stenting, OTSC^®^, use of cardiac septal defect occluders (CSDO), and rarely use of fibrin glue.

#### 3.2.2. Stenting

Stenting can be either esophageal, bronchial, or both [46,47]. Those stents can be fully or partially covered [48]. Esophageal stenting should solely be performed when there is no concomitant airway stenosis, the esophageal stent is not extrinsically compressing the airway, or the defect’s width is less than 20 mm. Otherwise, double stenting is recommended [49].

#### 3.2.3. OTSC^®^

OTSC^®^ was also used to close ERFs [50,51]. So far, publications have been limited to small series and mainly case reports. The results are very variable, with high direct success rates and short follow ups. Because of the lack of large cohorts, real conclusions about the effectiveness of treating ERF with OTSC^®^ remain unclear [48].

#### 3.2.4. CSDO

Invented in 1958 to treat atrial septal defects and composed of nitinol and polyester, this device has a shape memory used as a self-expanding double-disc device. Its thick waist portion serves to self-center the device during deployment to close the defect. The disc diameter varies from 9 to 54 mm, and the waist size varies from 4 to 38 mm. The nitinol and polyester play a major role in promoting closure and tissue ingrowth. Their off-label use to treat gastrointestinal fistulas was published in a systematic review in 2018. They showed a 100% technical success rate, 77.27% clinical success rate, and 22.72% adverse event rate, with no death related [52]. Li et al. reported the successful use of CSDO in six patients, with a direct closure rate of 100% [53].

#### 3.2.5. Surgical

Wang et al. primarily recommended the surgical approach to remove the fistula and the surrounding inflammatory tissue as a first choice in the treatment of ERF in their national expert consensus [46]. Depending on the severity of the case, a lobectomy or even a pneumectomy might be indicated. If a reanastomosis of the neoesophagus is not feasible, an esophageal diversion procedure will be performed. The bronchial defect is then closed by two layer-sutures followed by the insertion of a muscular flap between the defected structure of the bronchial tree and the esophagus. Structures such as the latissimus dorsi muscle, the major pectoral muscle, the pericard, and the pedicled intercostal muscle have been reported as anatomical structures suitable for a flap [54,55]. Surgery should always be considered as a salvage treatment after a failed endoscopic approach. In a retrospective study published in 2016, Lenz et al. concluded that patients eligible for surgical treatment had a better outcome compared to endoscopically treated patients (mainly with stents), although surgical candidates were the minority [56].

## 4. Chylothorax

The lymphatic fluid (almost 4 L/24 h) flows from the lower extremities and abdominal organs back to the brachiocephalic vein through the cisterna chyli at the level of the second lumbar vertebrae lateral to the aorta, where the thoracic duct that crosses to the left side of the body arises at the level of the fifth thoracic vertebrae. Because of its close relationship to the thoracic esophagus, an injury can occur during an esophagectomy. This happens in 5% of the cases after esophagectomy and can be caused by the direct injury of the thoracic duct or one of its branches, by a direct transection, by mass ligation, or while debulking a big tumor. Among the risk factors are a difficult lymphadenectomy regardless of the surgical approach (open or minimal invasive) [42], a lower body mass index [57,58], high intraoperative fluid balance, and neoadjuvant chemoradiotherapy [58,59]. CT is associated with malnutrition, prolonged pleural effusion, secondary pneumonia, and empyema. It has a high mortality rate, reaching 17% [60]. CT is divided according to the ECCG group depending on its treatment into Type I with enteric dietary modifications, Type II with total parenteral nutrition, and Type III with interventional or surgical therapy. It is also classified as low or high output depending on the amount of drained lymphatic fluid, respectively, below or over 1000 mL/day [12].

### 4.1. Prevention

The main method for prevention is preemptive thoracic duct ligation at the time of surgery. In a monocentric study of over 296 patients, a selective en masse ligation of the thoracic duct has been shown to be effective after preoperative oral intake of 120 mL olive oil for better detection of the chyle leak [61]. However, a systematic review and meta-analysis published in 2017 analyzed the data of seven comparative studies between prophylactic and non-prophylactic thoracic duct ligation in a total of 5226 Patients and showed no evidence of lowering the incidence of postoperative CT [62]. At our institution, thoracic duct ligation is not routinely performed.

### 4.2. Diagnosis

The diagnosis is easily made due to the classical milky appearance and excessive drainage through chest tubes usually after enteral nutrition [63]. A daily thorax drainage discharge of more than 10 mL/kg of body weight should be investigated [64]. The diagnosis is confirmed by a triglyceride level more than 110 mg/dL or the presence of chylomicrons in the effluent [65].

### 4.3. Therapy

Therapy depends on the daily output and the patient’s nutritional status. The optimal therapy is still debatable among conservative therapy, lymphangiography and embolization of the cisterna chyli, and surgical reintervention.

#### 4.3.1. Conservative Therapy

Once CT is diagnosed, the patient should be put on a restricted fat intake diet and total parenteral nutrition. Because of the fact that medium-chain triglycerides are directly absorbed through the intestinal mucosa into the portal vein, a diet containing medium-chain triglycerides is considered to provide the required daily fat intake without increasing the chyle leak output [66]. Alongside this dietary restriction, it is essential to monitor the pleural drainage output and to subcutaneously apply somatostatin analogs such as Octreotide 100 micrograms 3× daily. Negative predictive factors for the success of the conservative treatment are high-output CT by the time of diagnosis or a lack of improvement by the second day of therapy [67].

#### 4.3.2. Lymphangiography and Embolization of the Cisterna Chyli

First described by Cope et al. in 1997 [68], this procedure was accepted as the treatment of choice after a failed CT conservative therapy. It consists of three steps: lymphangiography, thoracic duct cannulation, and the subsequent lipiodol embolization. The lipiodol induces an inflammatory reaction at the site of lymphatic outflow, after aqueous hydrolysis and saponification of the surrounding fat tissue resulting in leak closure. This approach is considered rapid and technically easy, with success rates between 70 and 95% [60,69]. The risk of toxicity and a symptomatic pulmonary embolization should be taken into consideration when the cumulative dosage of lipiodol exceeds 20 mL [70,71].

#### 4.3.3. Surgery

Surgery is indicated in the case of a persistent high output CT after 5 days of conservative management or after failure of the radiological intervention. The aim of this approach is to identify the leak’s site and to ligate it. For this purpose, substances such as butter and cream, or methylene blue are preoperatively administered to help with the visualization of the leak. According to a recent systematic review published in 2021, the surgical approach was successful in 80% of the cases and was associated with a mortality rate between 0–33% and a morbidity rate of 6–33% [72].

## 5. Discussion

Esophagectomy with two fields of lymphadenectomy is the standard of care for resectable esophagus cancer. However, this surgery is associated with high risks of morbidity and mortality. The concept of centralization of care improved the results, reduced complication and death rates, and increased both short- and long-survival rates for several gastrointestinal malignancies including esophagus cancer [73]. An essential requirement for such centralization is the presence of well-trained, multidisciplinary teams of surgeons, radiologists, and endoscopists, who are available around the clock [23,73,74]. While treating complications, starting an adequate intervention earlier than 24 h after suspecting the diagnosis is associated with a substantial decrease in mortality, hospital stay, and treatment duration [75]. This is crucial for patient’s outcome.

A well-perfused anastomosis is a must for preventing complications, mainly AL and ERF. As mentioned above, a narrower conduit was believed to negatively affect vascularization. This aspect was studied by Zhang et al. and Tabira et al. and was found to have no impact on the rate of AL [76,77,78]. Intraoperatively, indocyanine green (ICG) fluorescence imaging devices are commonly used to test the anastomotic perfusion. These devices can clearly image the flow of ICG bound to proteins [79], but their main limitations are the inability of measuring the hemoglobin concentration and the tissue’s oxygen saturation in the anastomotic area. This could be overcome by using spectral novel techniques [80]. In a prospective comparative study, Sdralis et al. tested the prophylactic intraoperative use of a fibrin sealant to reinforce the anastomosis in a group of 57 patients. This did not show any benefit in lowering the AL rates [81]. Surgical revision was the traditional treatment of choice of AL to control sepsis and to reduce the fatal consequences of an ongoing mediastinitis. However, it was associated with a mortality rate reaching 40% [82].

In the last two decades, we observed a shift to non-operative approaches that gained popularity thanks to the advancements mainly in the endoscopic field. A review of the literature concerning the treatment modalities of AL after esophagectomy published in 2019 examined the results of 19 retrospective cohort studies including 273 patients and concluded that, due to the poor methodology, heterogenicity, and lack of AL definition, no particular treatment modality could be seen as superior to others. Depending on the treatment, the mortality rates were as follows: conservative (14%), endoscopic stent (8%), endoscopic vacuum-assisted closure system (0%), and surgery treatment group (50%) [83]. A recent monocentric retrospective study of 61 patients found that the conservative treatment was successful without the need of an added invasive procedure in 75% of the cases [21]. The fibrin glue method cannot be empirically advised because of the small sample sizes and limited studies and case reports on its application [81].

According to the guidelines of the European Society of Gastrointestinal Endoscopy (ESGE), clips should only be applied in clinically stable patients with perforations <2 cm and no signs of general inflammation in patients with esophageal perforations [84]. Lazar et al. summarized the available data and case reports. If TTSC was applied in patients with an esophageal perforation, an effective closure was reported in 88.8% (24/27). In patients treated with an OTSC^®^, successful healing was achieved in 92.8% (26/28) with no statistically significant difference (*p* = 0.12) but in very small group of patients [85]. One pitfall of this latter clip is the arduous removal once falsely released, which can be performed by lasers or cooling. [86]. Overall, the clinical success of treatment using the OTSC^®^ ranges from 63% to 89% according to several retrospective multicenter series with clinical success defined as closure of the gastrointestinal wall defect [87].

An endoscopic stent can be used when the AL defect is larger than 2 cm. While plastic stents are generally used for benign diseases, SEMSs have been used since the 1990s for long-term cancer related stenosis. In the following years, they were also implemented to treat esophageal perforations [88]. Fully covered SEMSs cannot have water-tight sealing of the leak even when totally expanded, which is why partially covered stents are more adequate specially in the case of ERF. The growing tissue around the stent will avoid the migration and the liquid flow from above [89]. That is why some authors prefer to use them more than the totally covered SEMSs. For others, this problem can be solved by fixing the upper edge of the stent using an OTSC^®^, at the same time avoiding tissue ingrowth and overgrowth and higher risk of bleeding at the time of extraction, which is more associated with partially covered stents. The overall success of endoscopic stenting was reported to be 81–86%, and it was generally recommended that the stent should be removed after 6–8 weeks of therapy [90,91,92,93,94,95,96]. The risk of failure of stent implantation increased in septic patients with a defect of >1.5 cm that had persisted for several weeks [94]. In some animal experiments, complete healing was only observed 4 weeks after the initial stent application [90,93].

Stent-associated mortality and complications were 10–13% and 13–34%, respectively [90,91,93,95]. Surgical intervention was needed to mainly deal with stent-related complications or incomplete healing of the initial injury only in 13% of cases [91,95]. Adverse events following stenting were dislocation 2–3% and migration 18–31% [90,91,93,94,97]. Long-term stent complications are secondary strictures and fistulization, associated upper-gastrointestinal hemorrhage [98], patient’s discomfort [95,99], aspiration, and reflux. Many options were described to minimize stent dislocation or migration. The stent can be attached by endoscopic clipping or fixed by suturing or heat sealing to the mucosa using an argon plasma beam. In the case of treatment failure, other options such as surgical intervention should be discussed. All decision making should be performed by a multidisciplinary team.

A recent systematic review of the literature and meta-analysis comparing EVT to SEMS for treating upper gastrointestinal tract mural defects showed that EVT promotes 21% improvement in complete fistula closure compared with the SEMS group (*p* = 0.0003). EVT demonstrated a 12% reduction in mortality rate (*p* = 0.0003), a 2-week shorter therapy duration (*p* = 0.00001), and 24% reduction in adverse events (*p* = 0.0001) [100]. While patients with EVT treatment are unable to be orally fed unless the sponge is extraluminally placed, patients treated with stents maintain their oral feeding uptake. For this matter, a combination of both techniques, the VACStent, was first reported by Chon et al. in 2019 in a case of AL after total gastrectomy. The patient benefited from the advances of the EVT and, at the same time, was able to be orally enterally fed thanks to the in situ stent [101]. In 2017, Neumann et al. published their series of treating six patients with ischemic mucosa after esophagectomy using the concept of pre-emptive EVT therapy. In 75% of the cases, the mucosa regenerated and no other procedure than EVT was necessary [102]. Gubler described the definition of this prophylactical use of EVT in 2018. The sponge was inserted at the time of the operation and left in situ for 3–5 days with a vacuum of 75 mm Hg. The results did not show any side effects and were not associated with higher morbidity or mortality rates [103]. As a consequence, this working group started a randomized controlled multicenter trial (the preSponge Trial) in 2021 [104]. Halvax et al. published data in which they demonstrated adequate and sufficient closure of transmural defects of up to 3 cm at a rate of 100%, using an endoluminal full-thickness suture in an animal model [105].

Case report series publications described the usage of the OverStitch system to close transmural perforation and postoperative AL and to fix SEMS in order to overcome their migration [106,107,108,109]. When reviewing data about the ERF, a lot of ambiguity surrounds the nomenclature of this type of fistula with the usage of the following terms to describe the same problem: tracheoesophageal or bronchoesophageal fistula [43], airway–gastric fistula [110], and esophagoalveolar and esophagopulmonary fistula [46]. A systematic review published in 2018 on the management of ERF after esophagectomy highlighted that while non-surgical managements are feasible and easy, they are usually associated with a higher failure rate when compared to the definitive surgical treatment [111]. Due to the fibrotic nature of fistula edges, a treatment with OTSC^®^ usually fails [112].

Miao et al. found that SCC was an independent predictor of CT, after reviewing 1290 consecutive cases of esophagectomy. He concluded, in this group of patients, a prophylactic ligation of the thoracic duct can be justified [57].

Data on the management of post esophagectomy complications are, to a large scale, mainly collected from small series and cases reports. Surgeon and center’s experience are key when it comes to treating AL, ERF, and CT.

## 6. Conclusions

Esophagectomy for cancer remains a complex surgery. Postoperative complications are common. To lower their rates complications, some pre-, intra- and postoperative measures should be taken. Clinical suspicion and liberal use of flexible endoscopy are essential to early diagnosis. Nowadays, nonoperative and minimally invasive treatments are widely available, and surgery is only indicated after the failure of previously tried approaches. Despite the lack of guidelines and consensus, there is no doubt that advances in endoscopy and radiology and the shift toward minimally invasive surgery with the centralization of care have left positive fingerprints in the management of complications after esophagectomy and, consequently, on the overall patient’s outcome.

## Figures and Tables

**Figure 1 cancers-14-05556-f001:**
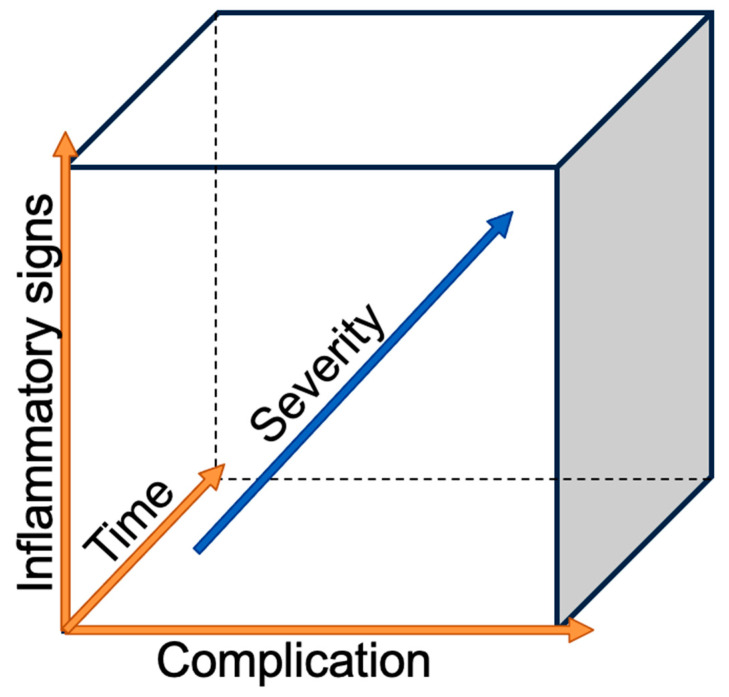
Severity of the complication in relation to the time and the inflammatory reaction.

**Figure 2 cancers-14-05556-f002:**
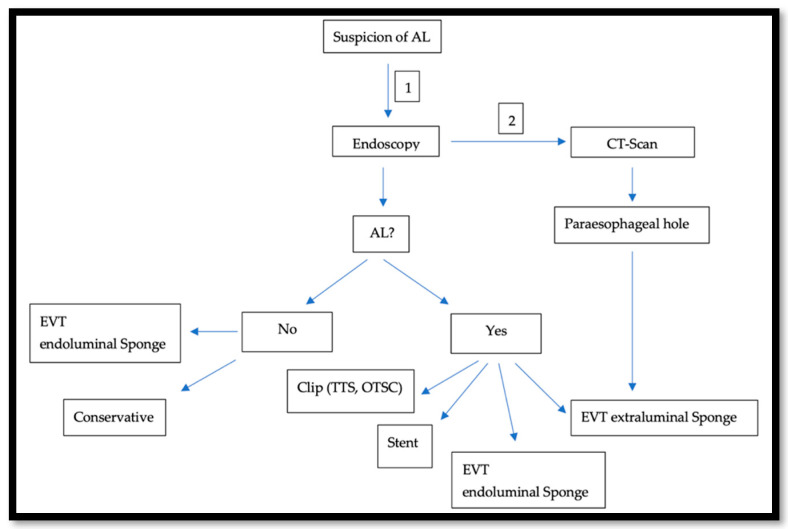
Illustrates our approach when an AL is suspected: Endoscopy is first performed (1) and is usually followed by a CT-Scan (2).

## Data Availability

Not applicable.

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
