# Peer review of "Adequate Management of Postoperative Complications after Esophagectomy: A Cornerstone for a Positive Outcome"

_cancers, 2022, doi:10.3390/cancers14225556_

Round 1
Reviewer 1 Report
Dear authors,
I want to congratulate you for an interesting and comprehensive manuscript. In my opinion you have done a great job summarizing the broad information about this topic.
However, the paper, although well written, has a discussion section imposible to read. I suggest dividing the information in different paragraphs to make it easily accesible.
Author Response
Dear Reviewer 1,
Thank you so much for taking your time to read and review our work. This was an excellent observation. We totally agree, the discussion section is a little dense. Following your suggestions, we divided the informations into different paragraphs, hoping to be more structured and clear for the readers.

Reviewer 2 Report
This review article describes systematically perioperative treatment and prevention for complications.
In my opinion, the contents of this article are close to perfect and there is no room for correction or supplementation. But, I propose to make corrections, these are very trivial.
Minor correction
Line 33 AC => Adenocarcinoma (AC)
Line 35 SCC => squamous cell carcinoma (SCC)
Line 51 AL => anastomotic leak (AL)
Line 51 ERF => esophago-respiratory fistula (ERF)
Line 51 CT => chylothorax (CT)
Line 198 esophago-respiratory fistula (ERF) => ERF
Line 268 Chylothorax => CT
Line 269 Chylothorax => CT
Line 282 Chylothorax => CT
Line 295 Chylothorax => CT
Line 302 Chylothorax => CT
Line 306 Chylothorax => CT
Line 314 Chylothorax => CT
Line 395 p=0,0003 => p=0.0003
Line 396 p<0,00001 => p=0.00001
Line 397 p<0,0001 => p=0.0001
Author Response
Dear Reviewer 2,
thank you very much for taking your time to read and review our work.
All minor errors have been corrected accordingly as you suggested in your report.

Reviewer 3 Report
Adequate Management of Postoperative Complications after Esophagectomy:A Cornerstone for a Positive Outcome.
The present manuscript provides the treating team with a very informative and excellent roadmap for the early detection and appropriate management of postoperative complications, particularly anastomotic leak, esophago-respiratory fistula and chylothorax.
For the three relevant complications, all diagnostic and therapeutic options are presented in a very clear manner and supported by plenty of literature data.
Precisely because of the lack of unanimous opinion and guidelines on this important topic, I see this work as a and concise and valuable recommendation on how to deal with complications after esophagectomy.
It is emphasized that a low threshold for checking their occurrence can be life-saving.
This manuscript should be published, even as it is
Author Response
Dear Reviewer 3,
Thank you very much for taking your precious time to read and review our work. We are very happy to read your nice comments and notes.